# EFFICIENT HYPERDIMENSIONAL COMPUTING

## ABSTRACT

Hyperdimensional computing (HDC) uses binary vectors of high dimensions to perform classification. Due to its simplicity and massive parallelism, HDC can be highly energy-efficient and well-suited for resource-constrained platforms. However, in trading off orthogonality with efficiency, hypervectors may use tens of thousands of dimensions. In this paper, we will examine the necessity for such high dimensions. In particular, we give a detailed theoretical analysis of the relationship among dimensions of hypervectors, accuracy, and orthogonality. The main conclusion of this study is that a much lower dimension, typically less than 100, can also achieve similar or even higher detecting accuracy compared with other state-of-the-art HDC models. Based on this insight, we propose a suite of novel techniques to build HDC models that use binary hypervectors of dimensions that are orders of magnitude smaller than those found in the state-of-the-art HDC models, yet yield equivalent or even improved accuracy and efficiency[1]. For image classification, we achieved an HDC accuracy of 96.88% with a dimension of only 32 on the MNIST dataset. We further explore our methods on more complex datasets like CIFAR-10 and show the limits of HDC computing.

## 1  INTRODUCTION

*Hyperdimensional computing* (HDC) is an emerging learning paradigm inspired by an abstract representation of neuron activity in the human brain using high-dimensional binary vectors. Compared with other well-known training methods like artificial neural networks (ANNs), HDCs have the advantage of high parallelism and low energy consumption (low latency). This makes HDCs well suited to resource-constrained applications such as electroencephalogram detection, robotics, language recognition and federated learning (Hsieh et al., 2021; Asgarinejad et al., 2020; Neubert et al., 2019; Rahimi et al., 2016). HDCs are also easy to implement in hardware (Schmuck et al., 2019; Salamat et al., 2019).

Unfortunately, the practical deployment of HDC suffers from low model accuracy and is always restricted to small and simple datasets. To solve the problem, one commonly used technique is increasing the hypervector dimension (Neubert et al., 2019; Schlegel et al., 2022; Yu et al., 2022). For example, running on the MNIST dataset, hypervector dimensions of 10,000 are often used. Duan et al. (2022) and Yu et al. (2022) achieved the state-of-the-art accuracies of 94.74% and 95.4% separately this way. In these and other state-of-the-art HDC works, hypervectors are randomly drawn from the hyperspace $\{-1, +1\}^d$, where the dimension $d$ is very high. This ensures high orthogonality, making the hypervectors more independent and easier to distinguish from each other (Thomas et al., 2020). As a result, accuracy is improved and more complex application scenarios can be targeted. However, the price paid due to higher dimension is in higher energy consumption possibly negating the advantage of HDC altogether (Neubert et al., 2019). This paper addresses this tradeoff.

In this paper, we will analyze the relationship between hypervector dimension and accuracy, as well as between dimension and orthogonality. In our analysis, we found that strict orthogonality can be obtained for small $d$. We will show that a dimension $d$ of only $2^{\lceil \log_2 n \rceil}$ is sufficient to yield $n$ vectors in $\{-1, 1\}^d$ with strict orthogonality. Dimensions higher than that are not necessary. If we relax orthogonality to $\varepsilon$-*quasi-orthogonality* (Kainen & Krkova, 2020), we will show that it is even easier to construct the hypervectors. Further, it is intuitively true that high dimensions will lead to high orthogonality (Thomas et al., 2020), contrary to popular belief, we found that as the

---

[1]https://anonymous.4open.science/r/LowHDC-F74B/README.md

dimension of the hypervectors $d$ increases, the upper bound for inference accuracy actually decreases (Statement 3.1 and Statement 3.2). In particular, if the hypervector dimension $d$ is sufficient to represent a vector with $K$ classes ($d > \log_2 K$) then, **the lower the dimension, the higher the accuracy.** The key insight of our work is this: *In HDC, it is not the higher dimension, that is the determinant of accuracy, and the required orthogonality for a given problem can be achieved at lower hypervector dimensions using our proposed techniques.*

Based on the analysis, we propose a combination of a novel trainable binary kernel-based encoder with the majority rule (shown in Figure 3) to reduce the hypervector dimension significantly while maintaining state-of-art accuracies. Running on the MNIST dataset, HDC accuracies of 96.88/97.23% were achieved with hypervector dimensions of only 32/64. The total number of calculation operations of our method is a mere 7% of the previous state-of-art related works where hypervectors dimensions of 10,000 or more were needed. We further explored our methods on CIFAR-10 and an HDC accuracy of 46.18% was achieved. Both our analysis and experiments show that dimensions of 5,000 or even 10,000 used by the state-of-the-art in HDC are not necessary.

The contribution of this paper is as follows:

- We give a comprehensive analysis of the relationship between hypervector dimension and the accuracy of HDC. Both the worst-case and average-case accuracy are studied. Mathematically, we explain why relatively lower dimensions can yield higher model accuracies. This contradicts the standard assumption in HDC. Furthermore, the relationship between orthogonality and hypervector dimension is also discussed. Based on the analysis, we can reduce the dimension by nearly three orders of magnitude.
- We introduce a kernel-based binary encoder and two HDC retraining algorithms. With these techniques, we can achieve higher detection accuracies using much smaller hypervector dimensions (latency) and better orthogonality compared to the state-of-the-art.

**Organisation** This paper is organized as follows. First, the basic workflow and background of HDC are introduced. Then, we describe our main dimension-accuracy and dimension-orthogonality analysis in Section 3. In Section 4, we present a trainable binary encoder and two HDC retraining approaches to improve accuracy while at the same time reducing energy consumption. We then show our experimental results and comparison with state-of-the-art HDCs in Section 5, followed by a discussion and conclusion.

## 2 BACKGROUND

Hyperdimensional computing encodes binary hypervectors with typical dimensions of 5,000 to 10,000 to represent the data. Using the MNIST dataset as an example, HDC encodes one float32-type image $f = f_0, f_1, ..., f_{783}$ to hypervectors by binding and adding the value hypervectors $\boldsymbol{v}$ and position hypervectors $\boldsymbol{p}$ together. Both these two hypervectors are independently drawn from the hyperspace $\{-1, +1\}^d$ randomly. Mathematically, we can construct representation $r$ for each image as followed:

$$r = \text{sgn}\left(\left(v_{f_0}\bigotimes p_{f_0} + v_{f_1}\bigotimes p_{f_1} + ... + v_{f_{783}}\bigotimes p_{f_{783}}\right)\right),$$

where $\text{sgn}(\cdot)$ is the sign function that binarizes the sum of hypervectors and returns -1 or 1. $\text{sgn}(0)$ is randomly assigned to 1 or -1. $\bigotimes$ is the *binding operation* that perform coordinate-wise (element-wise) multiplication. For example, $[-1, 1, 1, -1]\bigotimes[1, 1, 1, -1] = [-1, 1, 1, 1]$.

For training, all hypervectors $r_1, ..., r_{60,000}$ that of the same digit are added together. The *majority rule* is then used to generate the representation $R_c$ for class $c$

$$R_c = \text{sgn}\left(\sum_{i\in c} r_i\right). \tag{1}$$

For inference, the encoded test image is compared with the representation of each class $R_c$, and the most similar one is selected. Cosine similarity, L2 distance, and Hamming distance are commonly

used similarity measures in previous works. According to Frady et al. (2021), the inner product has the same function with Hamming distance for binary hyper vectors with values of -1 and 1, which we used in this work. The workflow is shown in the Appendix A.3.

## 3 HIGH DIMENSIONS ARE NOT NECESSARY

Compared to traditional ANNs, the use of binary vectors and simple, point-wise computation in HDC holds the promise of low energy consumption while achieving competitive accuracies. The Achilles Heel is in the high dimensions needed that potentially negated the gains. In this section, we will study the need for high hypervector dimensions in terms of both accuracy and orthogonality.

Through an analysis of the relationship between dimension and accuracy, we will show that a higher hypervector dimension does not necessarily lead to higher accuracy. We will show that for a classification task that has only two classes, a higher hypervector dimension results in both lower worst-case and average-case accuracy.

We then study the relationship between dimension and orthogonality and show that good orthogonality does not require high dimensions. This opens the door to performing HDC with significantly lower dimension hypervectors.

### 3.1 DIMENSION-ACCURACY ANALYSIS

To simplify the analysis of the HDC, we consider the following assumptions of hypervectors without loss of generality. We assume that the hypervectors are uniformly distributed over a $d$-dimensional unit ball:

$$\mathcal{X} = \{x \in \mathbb{R}^d \big| \|x\|_2 \leq 1\}.$$

Moreover, we assume that hypervectors $x$ are *linearly separable* and each class with label $i$ can be represented by $C_i$:

$$C_i = \{x \in \mathcal{X} | \theta_i \cdot x > \theta_j \cdot x, j \neq i\}, \quad 1 \leq i \leq K$$

where $\theta_i \in [0,1]^d$ are support hypervectors that are used to distinguish classes $i$ from other classes. This is a reasonable assumption as long as we select $d$ sufficiently large so that there exists a mapping (encoder) to embed the raw data into a $d$-dimensional unit ball.

Similarly, we define the prediction class $\hat{C}_i$ by $\hat{\theta}_i$ as followed:

$$\hat{C}_i = \{x \in \mathcal{X} | \hat{\theta}_i \cdot x > \hat{\theta}_j \cdot x, j \neq i\}, \quad 1 \leq i \leq K.$$

When we apply the majority rule to separate the above hypervectors $x$, we are approximating $\theta_i$ with $\hat{\theta}_i$ in the sense of maximizing the prediction accuracy. Here each $\hat{\theta}_i \in \{0,1\}^d$ is a binary vector.

Therefore we define the worst-case $K$-classes prediction accuracy over hypervectors distribution $\mathcal{X}$ in the following expression:

$$Acc_{K,d}^w := \inf_{\theta_1,\theta_2,\ldots,\theta_K} \sup_{\hat{\theta}_1,\hat{\theta}_2,\ldots,\hat{\theta}_K} \mathbb{E}_x\left[\sum_{i=1}^K \prod_{j \neq i} \mathbf{1}_{\{\theta_i \cdot x > \theta_j \cdot x\}} \mathbf{1}_{\{\hat{\theta}_i \cdot x > \hat{\theta}_j \cdot x\}}\right].$$

**Statement 3.1** *Assume $K = 2$, as the dimension of the hypervectors $d$ increases, the worst-case prediction accuracy decreases with the following rate:*

$$Acc_{2,d}^w = 2 \inf_{\theta_1,\theta_2} \sup_{\hat{\theta}_1,\hat{\theta}_2} \mathbb{E}_x\left[\mathbf{1}_{\{\theta_1 \cdot x > \theta_2 \cdot x\}} \mathbf{1}_{\{\hat{\theta}_1 \cdot x > \hat{\theta}_2 \cdot x\}}\right]$$

$$= \inf_{\theta_1,\theta_2} \sup_{\hat{\theta}_1,\hat{\theta}_2}\left[1 - \frac{\arccos\left(\frac{(\theta_1-\theta_2)\cdot(\hat{\theta}_1-\hat{\theta}_2)}{\|\theta_1-\theta_2\|_2\|\hat{\theta}_1-\hat{\theta}_2\|_2}\right)}{\pi}\right]$$

$$= 1 - \frac{\arccos\left(\frac{1}{\sqrt{\sum_{j=1}^d(\sqrt{j}-\sqrt{j-1})^2}}\right)}{\pi} \to \frac{1}{2}, \qquad d \to \infty$$

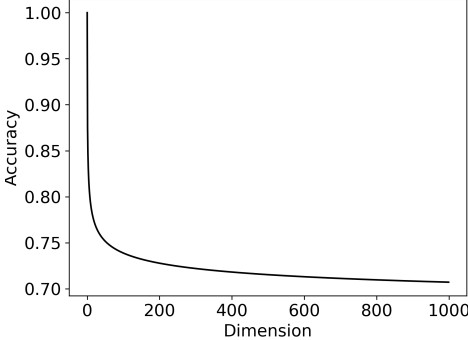
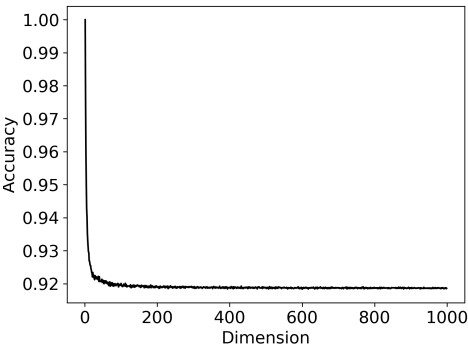

Figure 1: Worst-case Accuracy $Acc_{2,d}^w$ ——— Figure 2: Average-case Accuracy $\overline{Acc}_{2,d}$

*The first equality is by the symmetry of distribution $\mathcal{X}$. The second equality is the evaluation of expectation over $\mathcal{X}$ and the detail is given in Lemma A.1. For the third equality, the proof is given in Lemma A.3 and Lemma A.4.*

In the next statement, we further consider the average-case. Assume the prior distribution $\mathcal{P}$ for $\theta_1, ...\theta_K \sim \mathcal{U}[0,1]^d$. We can define the average accuracy in the following expression:

$$\overline{Acc}_{K,d} := \mathbb{E}_{\theta_1,\theta_2,...,\theta_K \sim \mathcal{P}} \sup_{\hat{\theta}_1,\hat{\theta}_2,...,\hat{\theta}_K} \mathbb{E}_x \left[ \sum_{i=1}^{K} \prod_{j \neq i} \mathbf{1}_{\{\theta_i \cdot x > \theta_j \cdot x\}} \mathbf{1}_{\{\hat{\theta}_i \cdot x > \hat{\theta}_j \cdot x\}} \right].$$

**Statement 3.2** *Assume $K = 2$, as the dimension of the hypervectors $d$ increases, the average case prediction accuracy decreases:*

$$\overline{Acc}_{K,d} = \mathbb{E}_{\theta_1,\theta_2 \sim U[0,1]^d} \sup_{\hat{\theta}_1,\hat{\theta}_2} \mathbb{E}_x \left[ \mathbf{1}_{\{\theta_1 \cdot x > \theta_2 \cdot x\}} \mathbf{1}_{\{\hat{\theta}_1 \cdot x > \hat{\theta}_2 \cdot x\}} \right]$$

$$= \mathbb{E}_{\theta_1,\theta_2 \sim U[0,1]^d} \sup_{\hat{\theta}_1,\hat{\theta}_2} \left[ 1 - \frac{\arccos\left( \frac{(\theta_1-\theta_2)\cdot(\hat{\theta}_1-\hat{\theta}_2)}{\|\theta_1-\theta_2\|_2\|\hat{\theta}_1-\hat{\theta}_2\|_2} \right)}{\pi} \right]$$

$$= \mathbb{E}_{\theta_1,\theta_2 \sim U[0,1]^d} \left[ 1 - \frac{\arccos\left( \sup_{j=1}^{d} \frac{\sum_{i=1}^{j}|\theta_1-\theta_2|_{(i)}}{\sqrt{j}\|\theta_1-\theta_2\|} \right)}{\pi} \right].$$

*Here $|\theta_1 - \theta_2|_{(i)}$ denotes the $i$-th maximum coordinate for vector $|\theta_1 - \theta_2|$.*

As the exact expression for the average-case accuracy is harder to evaluate, we do the Monte Carlo simulation which sampling $\theta_1$ and $\theta_2$ 1000 times to evaluate the expectation form. We then show the curve of $Acc_{K,d}^w$ and $\overline{Acc}_{K,d}$ over dimension from 1 to 1000 in Figure 1 and 2. It is easy to find that a high dimension for HDCs is not necessary for both the worst-case and average-case, the upper bound of accuracy will drop slowly when the dimension increases.

According to Tax & Duin (2002), we can approximate multi-class case where $K \geq 3$ by one-against-one binary classification. Therefore, we define the quasi-accuracy of $K$-class classification as follows:

$$Quasi\text{-}Acc_{K,d} = \frac{\sum_{i \neq j} Acc_{2,d}^{ij}}{K(K-1)},$$

where $Acc_{2,d}^{ij}$ can be either the average-case or worst-case accuracy that distinguishes class $i$ and $j$. Since the accuracy $Acc_{2,d}^{ij}$ for binary classification decreases as the dimension increase, the quasi-accuracy follows the same trend.

### 3.2 Dimension-orthogonality analysis

For strict orthogonality, we first construct a Hadamard matrix sequence $\{H_k\}$ (Horadam, 2012) as followed:

$$H_0 = [1];$$
$$H_1 = \begin{bmatrix} H_0 & H_0 \\ H_0 & -H_0 \end{bmatrix};$$
$$\vdots$$
$$H_k = \begin{bmatrix} H_{k-1} & H_{k-1} \\ H_{k-1} & -H_{k-1} \end{bmatrix}.$$

According to the definition of Hadamard matrix, if we take $n$ rows from $H_{\lceil \log_2 n \rceil}$, we can find $n$ hypervectors with strict orthogonality in $2^{\lceil \log_2 n \rceil}$-dimensional space. Then, we can give the following statement:

**Statement 3.3** *Dimension $d$ of only $2^{\lceil \log_2 n \rceil}$ is needed to find $n$ strictly orthogonal hypervectors, which indicates the unnecessity of high dimension.*

Further, if the Hadamard conjecture (Horadam, 2012) holds (for each positive integer $k$, there exists a Hadamard matrix of order $4k$), the number $d$ can be bounded above by $n + 3$.

We then consider the quasi-orthogonality since there is no enforcement of strict orthogonality in HDC's practice.

**Definition 3.1 ($\varepsilon-$quasiorthogonality)** *For two unit vectors $x$ and $y$, we call them quasi-orthogonal:*

$$|x^T y| \leq \varepsilon.$$

Based on the definition and recent progress on quasi-orthogonality (Kainen & Krkova, 2020) (shown in A.1), we can draw the following conclusion:

**Statement 3.4** *If orthogonality has been relaxed to $\varepsilon$-quasiorthogonality for $\varepsilon \in (0, 1)$, the number of $d$-dimensional vectors with $\varepsilon$-quasiorthogonality is exponential with respect to the dimension:*

$$n = O(e^{c(\varepsilon)d}).$$

*Here $c(\varepsilon)$ is a constant related to $\varepsilon$.*

Both Statement 3.3 and 3.4 indicate that even in the low dimension case, it is still feasible to find hypervectors with high (quasi-)orthogonality.

## 4 Methods

As shown in Figure 3, we first combine a *kernel-based binary encoder* with a fully-connected layer and train the whole network with cross-entropy loss function. Since the whole structure is binary, the *straight-through estimator* (STE) (Bengio et al., 2013) is used for back-propagation. Next, the fully-connected layer is replaced with a majority rule for power-saving. Weight sharing indicates that we use the same weights before and after we replacing the FC layer with the majority rule. Then, representations of each class $R_c$ can be obtained with Equation 1. To improve $R_c$, we train the combination of the binary encoder and majority rule with STE (Algorithm 2 Step 1) and recompute the final representations $R_c$ of each class $c$. Finally, all hypervectors are trained with Algorithm 2 Step 2 for higher detection accuracy.

### 4.1 Binary Kernel-based Encoder

The binary kernel-based encoder is composed of $k$ *binary neural netwrok* (BNN) style layers. Unlike the standard BNNs whose input is in floating point numbers, both the input and activation values

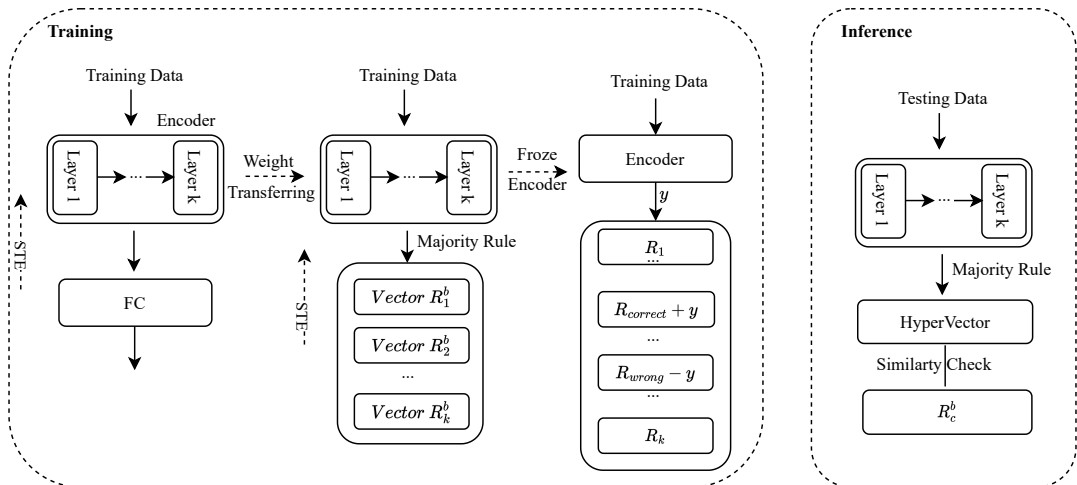

Figure 3: Workflow of Our HDC.

used in our structure have been quantized to 0 and 1. Since the information transmitted among layers is binary, we can replace the multiplication operations with addition operations. In particular, we only need to sum up the weights whose corresponding inputs are 1. Weights whose corresponding inputs are 0 can be ignored. For each neuron $i$ at layer $l$, if the sum up if higher than 1, a '1' is output. Otherwise, we output 0. Mathematically, :

$$x_i^l = \begin{cases} 0, & (\sum_{j=0}^n w_{i,j,x_j^{l-1}=1}^l + b_i^l) \leq 1 \\ 1, & (\sum_{j=0}^n w_{i,j,x_j^{l-1}=1}^l + b_i^l) > 1 \end{cases} \tag{2}$$

, where $x_i^l$ indicates the output of layer $l$ at neuron $i$, $w_{i,j}^l$ and $b_i^l$ indicates the weight and bias at layer $l$ ($j$ is the index of neurons in $l-1$ layer, and $i$ is the neuron index at $l$ th layer.), $w_{i,j,x_{l-1}=1}$ indicates the weight whose corresponding inputs $x_{l-1}$ are 1.

$G(x)$ is the gradient for the backpropagation. However, because the whole function is not continuous and not differentiable at the turning points, we use the method of the straight-through estimator to simulate the gradient. Thus $G(x)$ is set as 1 to make the whole network trainable: $G(x) \approx 1$.

After the training, we remove the fully connected layer and run the encoder to generate the binary representation $R_c^b$ of each class. The majority rule is used here, shown in Algorithm 1.

---

**Algorithm 1** representation Generation:

**Require:** $N$ number of training data $x$;
**Ensure:** Trained binary encoder $E$; Representation $R_c$ for class $c$ with dimension of $d$;Binary Representation $R_c^b$; Outputs of encoder $y$; Pre-defined Threshold $\theta$;
1: $y = E(x)$; $R_c = 0$
2: **for** $i = 1$ to $N$ **do**
3:     $R_c[i] += y_c$
4: **end for**
5: **for** $i = 1$ to $d$ **do**
6:     **if** $R_c[i] > \theta$ **then**
7:        $R_c^b[i] = 1$
8:     **else**
9:        $R_c^b[i] = 0$
10:     **end if**
11: **end for**

---

**Algorithm 2** HDC Retraining:

**Require:** Training data $x$ with label $R_c$; Trained Encoder $E$; $N$ training epochs.
1: **Step1:**
2: **for** epoch$= 1$ to $N$ **do**
3:     y $= E(x)$
4:     $L =$ mse(y, $R_c^b$) //Bp: STE
5: **end for**

6: **Step2:**
7: y $= E(x)$
8: **if** $y! = R_c^b$ **then**
9:     $R_{c_{correct}} += lr * y$
10:     $R_{c_{wrong}} -= lr * y$
11: **end if**
12: Generate $R_c^b$ (Algorithm 1, line 5-9)

---

## 4.2 RETRAINING

Here, we introduce a two-step retraining method. As shown in Algorithm 2, training data are first sent to the encoder in batches. The mean squared error is used as the loss function to update weights in the encoder. Then, we freeze the encoder and update the representation of each class. If the output $y$ is wrongly detected as class $c_{wrong}$ which should belong to class $c_{correct}$, we minus the representation of wrong class $R_{c_{wrong}}$ by the multiplication of learning rate and $y$. Meanwhile, we add the representation of the correct class $R_{c_{right}}$ by the multiplication of the learning rate and $y$ as well. Then, the modified $R_c$ are sent to Algorithm 1 to generate the binary representation $R_c^b$.

## 4.3 INFERENCE

As we have already computed the representation of each class, we can simply compare the similarity between the resulting hypervector (computed by sending the test data to the same encoder) and the representation of all classes. Then, we output the class with the highest similarity. We turn the value of 0 in $R_c^b$ to $-1$ and do the inner product for similarity check. Orthogonality of the resulting representation $\boldsymbol{R_c}$ can be evaluated with Equation 3. The closer $\bar{O}$ is to 0, the better the orthogonality.

$$\bar{O} = \frac{1}{K(K-1)} \sum_{c_1 \neq c_2} \frac{|\mathbf{R}_{c_1}^b * \mathbf{R}_{c_2}^{b\ T}|}{d} \tag{3}$$

## 5 RESULTS

We have implemented our schemes in CUDA-accelerated (CUDA 11.7) PyTorch version 1.13.0. The experiments were performed on an Intel Xeon E5-2680 server with two NVIDIA A100 Tensor Core GPUs and one GeForce RT 3090 GPU, running 64-bit Linux 5.15. MNIST dataset[2] and CIFAR10[3] are used in our experiments.

## 5.1 A CASE STUDY OF OUR TECHNOLOGIES

Here, we will describe how our approaches improve the digit recognition task step by step.

### 5.1.1 TRAINING THE ENCODER

We build a three-layer binary kernel-based encoder to enhance the HDC model. For each layer, we set the output channel number as 32, kernel size as 6, and stride as 2.

We first discuss the relationship between the pre-defined threshold mentioned in Algorithm 1 with accuracy. As shown in Figure 4, running on the MNIST dataset, taking the hypervector dimension of 16 as an example (full ablation studies are shown in the Appendix), we find that the threshold has good robustness against the noise. The detection accuracy remains almost the same when the threshold varies from 500 to 5000 (the max number in $R_c$ after encoder without majority rule is around 6500) and good orthogonality is also achieved.

We further consider the relationship between the dimension and inference accuracy with the most suitable threshold. As shown in Figure 5, we can achieve an HDC accuracy of 96.82/97.23% with a dimension of only 32 and 64. Also, the accuracy will drop when the dimension is higher than 128, which is consistent with Statement 3.1.

### 5.1.2 HDC RETRAINING

Thus far, we have shown how we can achieve the-state-of-art HDC accuracy with the smallest hypervector dimension. We can in fact improve the results using retraining techniques we will describe in this section. For example, with a dimension of 32, we can push the accuracy to 96.88% with our

---

[2]http://yann.lecun.com/exdb/mnist/
[3]https://www.cs.toronto.edu/ kriz/cifar.html

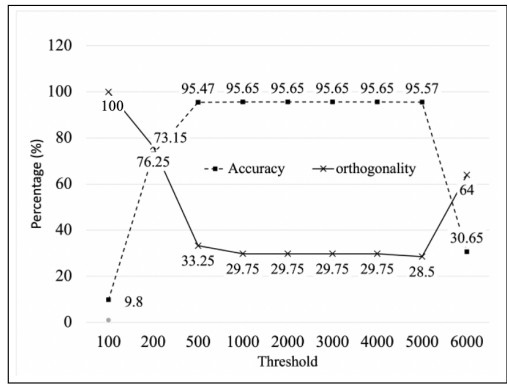 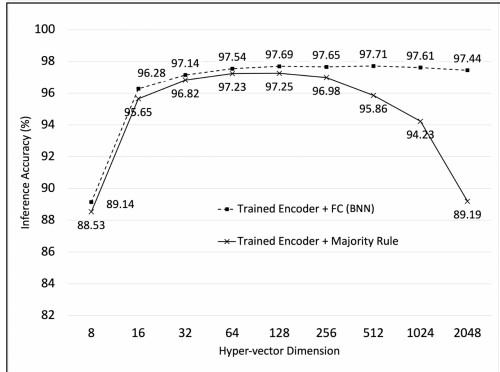

Figure 4: Threshold Study. The orthogonality is measured using Equation 3

Figure 5: Dimension Study

two-step training (0.05% and 0.01% accuracy improvement with steps 1 and 2, respectively). However, as shown in Figure 5, there is almost no accuracy drop after replacing the fully-connected layer with the majority rule, which indicates accuracy improvement after retraining may not be significant for the MNIST datasets. Therefore, we will explore our retraining methods on the CIFAR-10 dataset. Using the same hypervector dimension, a baseline accuracy (trained encoder+majority rule) of 38.42% was achieved. After retraining step 1 and step 2, the accuracy has improved by 0.21% and 0.42% respectively. The final accuracy increased to 39.05% in a matter of minutes.

## 5.2 EXPERIMENTAL RESULTS

Our full set of experiment results is shown in Table 1 where we compare our accuracy, dimension, and number of operations with other state-of-the-art HDC models. In this paper, HDC accuracies of 96.88% and 97.23% with $d = 32$ and $64$ were achieved for the MNIST dataset. We also applied our techniques to a larger dataset to test whether they work in more complex situations. For the CIFAR-10 dataset, an HDC accuracy of 39.05% with $d = 32$ was achieved with 1.17M computations. When we increase the dimension to 128, we can achieve an HDC accuracy of 46.18% with 4.34M computations.

A number of state-of-the-art HDC works were chosen for comparison. TD-HDC, proposed by Chuang et al. (2020), is a threshold-based framework to dynamically choose an execution path. They can improve the accuracy-energy efficiency trade-off and achieve an HDC accuracy of 88.92% on MNIST with their pure binary HD model. Hassan et al. (2021) used a basic HDC model on the MNIST dataset in their case study. They encoded the pixels based on their black/white value and used majority sum operation in the training stage to combine similar samples. They achieved an HDC accuracy of 86% on the MNIST dataset. HDC is also used in federated learning and secure learning. FL-HDC by Hsieh et al. (2021) focused on the combination of HDC and federated learning. They introduced the polarized model into the federated learning field to reduce communication costs and managed to control the accuracy drop by retraining. 88% accuracy was achieved on the MNIST dataset. SecureHD (Imani et al., 2019b) adapted a novel encoding and decoding method to perform securely learning tasks with the idea of HDC. Their accuracy on the MNIST dataset was 95% for federated training. More recently, LeHDC (Duan et al., 2022), by transferring the HDC classifier into the binary neural network, has used the learning-based HDC to achieve 94.74% on the MNIST dataset and 46.10% on the CIFAR-10 dataset. QuantHD (Imani et al., 2019a) and SearcHD (Imani et al., 2019c) are two methods that introduce multi-model and retraining into the HDC field. In LeHDC, they report the accuracy of the CIFAR-10 dataset with the methods of QuantHD and SearcHD as baselines, which are 22.66% and 28.42%. Compared with HDC, binary neural networks always require additional multiplication operations at least in the first layer because of the floating point input, which is much more expensive and was not considered in our comparison.

For inference, cosine similarity and Hamming distance are used in most state-of-the-art works. Since cosine distance requires additional multiplication and division operations which are quite expensive,

we chose Hamming distance instead. The number of operations in Hamming distance is linearly proportional to the dimension of the hypervectors, which indicates that our method only needs 0.32% operations compared with other HDCs with a dimension of 10,000.

Table 1: Comparison with related works.

| | Accuracy | Dimension | Inference | |
| | | | Encoder addition/Boolean op count | Similarity |
|---|---|---|---|---|
| | | MNIST | | |
| SearcHD | 84.43% | 10,000 | 7.84M/7.84M | Hamming |
| FL-HDC | 88% | 10,000 | 7.84M/7.84M | Cosine |
| TD-HDC | 88.92% | 5,000 | 3.92M/3.92M | Hamming |
| QuantHD | 89.28% | 10,000 | 7.84M/7.84M | Hamming |
| LeHDC | 94.74% | 10,000 | 7.84M/7.84M | Hamming |
| SecureHD | 95% | 10,000 | 7.84M/7.84M | Cosine |
| **This work** | **96.88%** | **32** | **1.15M/0** | Hamming |
| **This work** | **97.23%** | **64** | **1.19M/0** | Hamming |
| | | CIFAR-10 | | |
| SearcHD | 22.66% | 10,000 | 10.24M/10.24M | Hamming |
| QuantHD | 28.42% | 10,000 | 10.24M/10.24M | Hamming |
| LeHDC | 46.10% | 10,000 | 10.24M/10.24M | Hamming |
| **This work** | 39.05% | **32** | **1.17M/0** | Hamming |
| **This work** | 46.18% | **128** | **4.34M/0** | Hamming |

## 6 DISCUSSION

Our analysis of the relationship between orthogonality and the number of classes also affects the results. This has been largely ignored in other HDC works. Taking the MNIST dataset as an example, the state-of-the-art works use hypervectors of dimensions 5,000 to 10,000 to 'distinguish' pixel values that range from 0 to 255. However, if quantization is applied to input data and a proper encoder is used to extract the information from the original picture, theoretically, a much smaller dimension ($K$ reduced to 10 because the MNIST dataset has 10 labels) is needed. This also explains why our method and other HDCs cannot work on more complex datasets like ImageNet where the number of classes is large.

## 7 CONCLUSION

In this paper, we considered the dimension of the hypervectors used in HDC. We presented a detailed analysis of the relationship between dimension and accuracy as well as the relationship between dimension and orthogonality to demonstrate that it is not necessary to use high dimension to get a good performance in HDC. We showed that it is the orthogonality that affects accuracy. Previous works have been using high dimensions to achieve higher orthogonal because they used randomly drawn hypervectors. We showed that the orthogonality required to solve the problem can be achieved without resorting to high dimensions. As a result, we can reduce the dimensions from the tens of thousands used by the state-of-the-art to merely tens, while achieving the same level of accuracy. Computing operations during inference have been reduced to a tenth of that in traditional HDCs. Running on the MNIST dataset, we achieved an HDC accuracy of $96.88\%$ using a dimension of only 32. All our results are reproducible using the code we have made public.

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

# A APPENDIX

## A.1 ABLATION STUDY

We show the full threshold study with dimensions varied from 8 to 1024:

Figure 6: Threshold Study

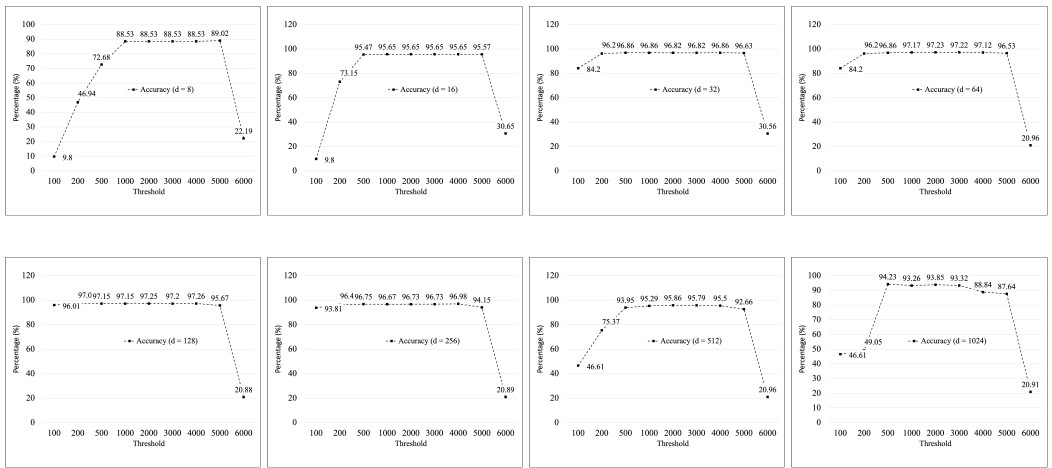

## A.2 LEMMA AND PROOF

**Lemma A.1**

$$\mathbb{E}_x\left[\mathbf{1}_{\{\theta_1\cdot x>\theta_2\cdot x\}}\mathbf{1}_{\{\hat{\theta}_1\cdot x>\hat{\theta}_2\cdot x\}}\right]=\frac{1}{2}(1-\frac{\arccos(\frac{(\theta_1-\theta_2)\cdot(\hat{\theta}_1-\hat{\theta}_2)}{\|\theta_1-\theta_2\|_2\|\hat{\theta}_1-\hat{\theta}_2\|_2})}{\pi}).$$

*Proof.* Consider the plane spanned by vector $\theta_1-\theta_2$ and $\hat{\theta}_1-\hat{\theta}_2$ and the projection of $x$ to this plane, the two indicator function requires the angle $<Px,\theta_1-\theta_2>$ and angle $<Px,\hat{\theta}_1-\hat{\theta}_2>$ to be smaller than $\frac{\pi}{2}$. Evaluating the expectation over $\mathcal{X}$ is equivalent to evaluating the intersection region of two semicircles. Therefore the result is $\frac{\pi-\arccos(\frac{(\theta_1-\theta_2)\cdot(\hat{\theta}_1-\hat{\theta}_2)}{\|\theta_1-\theta_2\|_2\|\hat{\theta}_1-\hat{\theta}_2\|_2})}{2\pi}$. □

**Lemma A.2** *When the coordinates of vector $\Delta\theta$ are ordered by absolute value:* $1\geq|\Delta\theta_1|\geq|\Delta\theta_2|\geq\cdots\geq|\Delta\theta_d|$. *Then we have the following equality:*

$$\sup_{\Delta\hat{\theta}\in\{-1,0,1\}^d}\frac{\Delta\theta\cdot\Delta\hat{\theta}}{\|\Delta\theta\|_2\|\Delta\hat{\theta}\|_2}=\sup_{1\leq j\leq d}\{\frac{\sum_{i=1}^{j}|\Delta\theta_j|}{\sqrt{j}\|\Delta\theta\|_2}\}.$$

*Proof.* By the definition of the supremum, iterate over the list $\Delta\hat{\theta}\in[\mathbf{e}_1,\mathbf{e}_1+\mathbf{e}_2,\ldots,\mathbf{e}_1+\mathbf{e}_2+\cdots+\mathbf{e}_d]$, $\mathbf{e}_i$ is the unit vector with the same sign as $\Delta\theta_i$, we know

$$\sup_{\Delta\hat{\theta}\in\{-1,0,1\}^d}\frac{\Delta\theta\cdot\Delta\hat{\theta}}{\|\Delta\theta\|_2\|\Delta\hat{\theta}\|_2}\geq\sup_{1\leq j\leq d}\{\frac{\sum_{i=1}^{j}|\Delta\theta_j|}{\sqrt{j}\|\Delta\theta\|_2}\}.$$

Now we show the $\leq$ part. We show that when the $\Delta\theta$'s coordinates are ordered, the optimal $\Delta\hat{\theta}$ is of the form

$$(\text{sign}(\Delta\theta_1),\ldots,\text{sign}(\Delta\theta_j),0,\ldots,0).$$

For any $\Delta\hat{\theta}$ with norm $\sqrt{j}$,

$$\Delta\theta\cdot\Delta\hat{\theta}\leq\sum_{i=1}^{j}|\Delta\theta_j|.$$

Therefore,

$$\sup_{\Delta\hat\theta\in\{-1,0,1\}^d}\frac{\Delta\theta\cdot\Delta\hat\theta}{\|\Delta\theta\|_2\|\Delta\hat\theta\|_2}=\sup_j\sup_{|\Delta\hat\theta|=\sqrt{j},}\frac{\Delta\theta\cdot\Delta\hat\theta}{\|\Delta\theta\|_2\|\Delta\hat\theta\|_2}\le\sup_{1\le j\le d}\{\frac{\sum_{i=1}^j|\Delta\theta_j|}{\sqrt{j}\|\Delta\theta\|_2}\}.$$

$\square$

**Lemma A.3**

$$\inf_{\theta_1,\theta_2}\sup_{\hat\theta_1,\hat\theta_2}\left[1-\frac{\arccos(\frac{(\theta_1-\theta_2)\cdot(\hat\theta_1-\hat\theta_2)}{\|\theta_1-\theta_2\|_2\|\hat\theta_1-\hat\theta_2\|_2})}{\pi}\right]\le 1-\frac{\arccos(\frac{1}{\sqrt{\sum_{j=1}^d(\sqrt{j}-\sqrt{j-1})^2}})}{\pi}.$$

*Proof.* We will show the $\le$ part by construction. Set $\theta_1=(1,\sqrt2-\sqrt1,\ldots,\sqrt d-\sqrt{d-1}),\theta_2=(0,0,\ldots,0)$. According to the Lemma A.2 and the monotonicity of $\arccos$ function, we have

$$\inf_{\theta_1,\theta_2}\sup_{\hat\theta_1,\hat\theta_2}\left[1-\frac{\arccos(\frac{(\theta_1-\theta_2)\cdot(\hat\theta_1-\hat\theta_2)}{\|\theta_1-\theta_2\|_2\|\hat\theta_1-\hat\theta_2\|_2})}{\pi}\right]\le\sup_{\hat\theta_1,\hat\theta_2}\left[1-\frac{\arccos(\frac{(\theta_1-\theta_2)\cdot(\hat\theta_1-\hat\theta_2)}{\|\theta_1-\theta_2\|_2\|\hat\theta_1-\hat\theta_2\|_2})}{\pi}\right]$$

$$=1-\frac{\arccos(\sup_{\hat\theta_1,\hat\theta_2}\frac{\theta_1-\theta_2)\cdot(\hat\theta_1-\hat\theta_2)}{|\theta_1-\theta_2||\hat\theta_1-\hat\theta_2)|})}{\pi}$$

$$=1-\frac{\arccos(\frac{1}{\sqrt{\sum_{j=1}^d(\sqrt{j}-\sqrt{j-1})^2}})}{\pi}.$$

$\square$

**Lemma A.4**

$$\inf_{\theta_1,\theta_2}\sup_{\hat\theta_1,\hat\theta_2}\left[1-\frac{\arccos(\frac{(\theta_1-\theta_2)\cdot(\hat\theta_1-\hat\theta_2)}{\|\theta_1-\theta_2\|_2\|\hat\theta_1-\hat\theta_2\|_2})}{\pi}\right]\ge 1-\frac{\arccos(\frac{1}{\sqrt{\sum_{j=1}^d(\sqrt{j}-\sqrt{j-1})^2}})}{\pi}.$$

*Proof.* If there exists $\theta_1^*,\theta_2^*$ such that the LHS is smaller than $1-\frac{\arccos(\frac{1}{\sqrt{\sum_{j=1}^d(\sqrt{j}-\sqrt{j-1})^2}})}{\pi}$, by monotonicity we know

$$C_0:=\sup_{\hat\theta_1,\hat\theta_2}\frac{(\theta_1^*-\theta_2^*)\cdot(\hat\theta_1-\hat\theta_2)}{|\theta_1^*-\theta_2^*||\hat\theta_1-\hat\theta_2|}<\frac{1}{\sqrt{\sum_{j=1}^d(\sqrt{j}-\sqrt{j-1})^2}}=:C_d.$$

Denote $\Delta\theta^*=\theta_1^*-\theta_2^*$. Without loss of generality, we assume $\Delta\theta^*\in[0,1]^d,\|\Delta\theta^*\|_2=1$ and

$$\Delta\theta_1^*\ge\Delta\theta_2^*\ge\cdots\ge\Delta\theta_d^*.$$

Starting from $\Delta\theta_1^*,\ldots,\Delta\theta_d^*$, we construct another feasible solution $\Delta\theta_1,\ldots,\Delta\theta_d$ without increasing the corresponding supremum value beyond $C_0$. However, if we compare $\Delta\theta_1,\ldots,\Delta\theta_d$ element-wisely with $(\sqrt k-\sqrt{k-1})C_0,1\le k\le d$, the first $\Delta\theta_k$ that is not equal to $(\sqrt k-\sqrt{k-1})C_0$ is greater than $(\sqrt k-\sqrt{k-1})C_0$, this gives us the contradiction to the $C_0$'s definition.

Also notice $\sum_{i=1}^d(\Delta\theta_i^*)^2=1,C_0<C_d$, there always exists index $k$ satisfying $\Delta\theta_k^*>(\sqrt k-\sqrt{k-1})C_0$.

Assume the first $\theta_i$ that is not equal to $(\sqrt i-\sqrt{i-1})C_0$ is still smaller than $(\sqrt i-\sqrt{i-1})C_0$. By the above paragraph, we can find the first $\theta_k,k>i$ with $\theta_k^*>(\sqrt k-\sqrt{k-1})C_0$. Then we adjust $(\theta_i^*,\theta_k^*)$ to $((\sqrt i-\sqrt{i-1})C_0,\sqrt{(\theta_i^*)^2+(\theta_k^*)^2-(\sqrt i-\sqrt{i-1})^2C_0^2})$. We can verify that the assumed inequalities continue to hold. (There are cases for $(\theta_i^*)^2+$

$(\theta_k^*)^2 - (\sqrt{i} - \sqrt{i-1})^2 C_0^2 \leq (\sqrt{k} - \sqrt{k-1})^2 C_0^2$, then we just end this modification with $(\sqrt{(\theta_i^*)^2 + (\theta_k^*)^2 - (\sqrt{k} - \sqrt{k-1})^2 C_0^2}, (\sqrt{k} - \sqrt{k-1}) C_0)$ and then repeat the procedure. )

As the above procedure repeats, number $\#\{k|\theta_k = (\sqrt{k} - \sqrt{k-1}) C_0\}$ is strictly increased. When it stopped, the first non-$(\sqrt{k} - \sqrt{k-1}) C_0$ term is larger than the $(\sqrt{k} - \sqrt{k-1}) C_0$ and this gives us the contradiction. $\qquad\square$

**Definition A.1 (Hadamard matrix)** *A Hadamard matrix is a square matrix with each entry being 1 or $-1$ and whose rows are mutually orthogonal.*

$$HH^T = dI_d.$$

**Theorem A.1 (Kainen & Krkova, 2020)** *The $\varepsilon$-quasiorthogonal dimension of $\mathbb{R}^d$,*

$$dim_\varepsilon(d) := \max\{|X| : X \subset S^{n-1}, x \neq y \in X \Rightarrow |x \cdot y| \leq \varepsilon\} \geq e^{d\varepsilon^2/2}.$$

### A.3 WORKFLOW OF TRADITIONAL HDCS

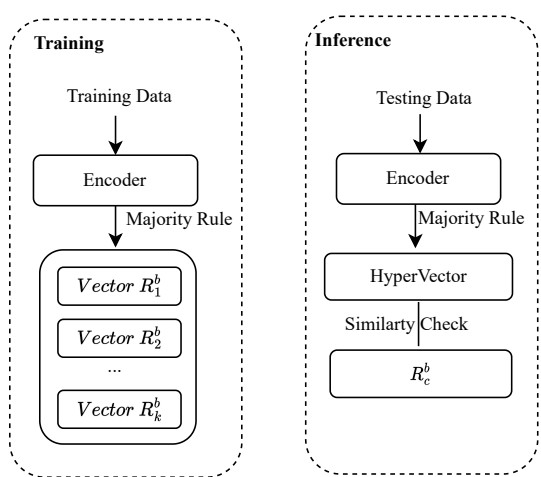

Figure 7: Workflow of Traditional HDC.

, where $R_c^b$ indicates the binary final representation of each class $c$.

### A.4 EXPLANATION OF THE LOW ACCURACY FOR LOW-DIMENSION ($d \leq 64$) IN NUMERICAL EXPERIMENTS

As can be seen from Figure 1, 2 and Figure 5, the current Statement 3.1 and 3.2 do not predict the low accuracy for dimension $d \leq 64$. This is caused by the breaking down of the assumption that data can be embedded in a $d$-dimensional linearly separable unit ball.

Consider a different setup in that the underlying dimension for data is fixed to be $m$. Each class is defined to be:

$$C_i = \{x \in \mathbb{B}^m | \theta_i \cdot x > \theta_j \cdot x, j \neq i\}, \quad 1 \leq i \leq K.$$

Assume that the linear projection of data from $m$-dimensional linearly separable unit ball to $d$-dimensional ($d < m$) space in a coordinate-wise approach. It is equivalent to optimizing over the following hypervector set

$$\Theta_{co_1,\ldots,co_d} = \{\theta | \theta_i \in \{0,1\}, i \in \{co_1,\ldots,co_d\}; \theta_i = 0, i \notin \{co_1,\ldots,co_d\}\},$$

Here $co_1,\ldots,co_d$ are the coordinates index of the projected space.

The worst-case $K$-classes prediction accuracy of the $m$-dimensional data projected onto a $d$-dimensional subspace is

$$
\begin{aligned}
Acc^w_{K,m,d} := \inf_{\theta_1,\theta_2,\ldots,\theta_K \in [0,1]^m} \sup_{co_1,\ldots,co_d} \sup_{\hat{\theta}_1,\hat{\theta}_2,\ldots,\hat{\theta}_K \in \Theta_{co_1,\ldots,co_d}} & \mathbb{E}_x \left[ \sum_{i=1}^K \prod_{j\neq i} \mathbf{1}_{\{\theta_i \cdot x > \theta_j \cdot x\}} \mathbf{1}_{\{\hat{\theta}_i \cdot x > \hat{\theta}_j \cdot x\}} \right] \\
& \leq Acc^w_{K,m,d+1} \\
& \leq Acc^w_{K,m}.
\end{aligned}
$$

The monotonicity comes from the fact that the two supremums are taken over a monotonic hyper-vector set $\Theta$ sequence.

When we randomly project the high-dimensional data into a low-dimensional space, the accuracy of the majority rule suffers from two types of error, namely, misclassification and misrepresentation.

Here the misclassification error refers to the error from approximating a $d$-dimensional linearly separable data while the misrepresentation error refers to the error caused by the difference between the projected distribution from the $m$-dimensional unit ball and a $d$-dimensional uniform distribution over the unit ball.

The misclassification error is characterized by the Statement 3.1 and 3.2. However, when the dimension $d$ is small, the misrepresentation error can be much larger. This explains the left part ($d \leq 64$) of the curves in our numerical experiments.

## A.5 ADDITIONAL NUMERICAL RESULTS

Here we provide the numerical results when the weights in encoder is changed from floats to integers. The results in Table 2 shows that performance does not degenerate when the integer weights are used in the encoder.

Table 2: Additional numerical results for integer encoder weights on MNIST.

| Dimension | Accuracy |
|---|---|
| **16** | **90.83%** |
| **32** | **95.12%** |
| **64** | **95.63%** |
| **128** | **95.73%** |

