# OpenReview forum: "Efficient Hyperdimensional Computing"
_ICLR.cc/2023/Conference — Submitted to ICLR 2023_

### Official Review · Reviewer_jTeX · 2022-10-24

**Confidence:** 4
**Correctness:** 2
**Technical Novelty And Significance:** 2
**Empirical Novelty And Significance:** 2
**Recommendation:** 5

**Clarity, Quality, Novelty And Reproducibility:**

The work is original, well-cited, however, some claims and empirically results are not well-supported.

**Strength And Weaknesses:**

This paper examines the necessity for high dimension in HDC, however, the reasoning can not convince me in the following aspects:

1. In standard HDC, hypervectors in the item memory is simply sampled from a high dimensional space, and the encoding process involves only binding and bundling, which are highly parallel and efficient operations. However, this paper extracts the hypervector for each data through training a binary neural network, thus the comparison in table 1 is not fair for some baselines which samples hypervectors, as the training/encoding cost is totally different, plus only inference complexity is given in table 1, not to say the retraining cost.

-- 1.1 Further more, this method nearly abandons the advantage of HDC, using binary NN to derive hypervectors, why not just use a simple BNN? One can imagine, use an even complicated network to derive useful features as hypervectors, follows with the HDC framework, it would have same inference cost as the features are also binarized. Therefore, a comparison with the same BNN used should be included as a baseline to justify the improvement is not from training NNs. It's not surprising to me that this method can perform well in a low dimension.

2. The author claims it's the orthogonality, not a high dimension, that is the determinant of HDC accuracy

-- 2.1 if that holds true, a simple experiment is that, as shown in section 3.2, you can find $n$ strictly orthogonal hyper vectors in a low dimension $2^{\lceil{\log_2(n)}\rceil}$ space by simply picking $n$ rows in the Hadamard matrix, then you can use the standard HDC encoding process for training, why suffer using a neural network to find hypervectors? Follow the author's logic, if this performs well, I would buy the claim, though I doubt it.
-- 2.2 even with a trained BNN, did you test that the learnt low dimensional hypervectors are orthogonal to each other? I doubt it.
-- 2.3 As the cited work Yu et al., 2022 shows, the orthogonality is not helpfully sometimes, but a related (with covariance) random sampled hypervectors help boost the HDC performance.
-- 2.4 The analysis in section 3 looks interesting, however the conclusion holds only when the hypervectors are sampled uniformly from the ball, but not for general HDC, so is the claim that "the lower the dimension, the higher the accuracy." to be conditioned, not for HDC. Though the result is still counter intuitive to me.
-- 2.5 The high dimension brings not only orthogonality, but also robustness, i.e., the hypervector can be recovered even a large fraction of its element perturbed, this property would lost in a low dimensional space.

3. After checking through the code, it looks to me the (MNIST) data to you BNN is floats? If not, how do you quantize the pixel values of images to binary?

**Summary Of The Paper:**

This paper uses a neural network ($k$ binary neural network layers) as an encoder to derive binary features and then plug into hyperdimensional computing framework for inference. The paper manages to arrive at satisfactory performance on MNIST with a low dimension as 32, however, the connection from the motivation and analysis to the result is not clear to me.

**Summary Of The Review:**

Overall, it is still an interesting submission, currently I think it's under the bar of ICLR, unless some suggested experiments can be conducted to support the claims in the paper.

---

> ### Author Response · Authors · 2022-11-10
> **Response to Reviewer jTeX**
>
> 1. Standard HDCs have three main components: the encoder that uses binding and bundling to generate hypervectors, a majority rule that generates the representation of each class, and a distance function that checks for similarity. For the encoder, as shown in Table 1, our proposal uses less addition and boolean operations compared to traditional HDCs. We believe this results in better energy efficiency. The training time of our encoder is short. For example, for MNIST with $d=16$ on MNIST, it only takes 28 seconds to achieve a good enough accuracy (in our experiments, traditional HDCs take minutes to train because of the large dimension). Since we use the same majority rule and distance function, we are just as parallel and efficient as traditional HDCs albeit with much smaller hypervector dimensions. As we mentioned in Section 5.2, compared with HDC, binary neural networks always require additional multiplication operations, at least in the first layer, because the input is in floating point. This is much more expensive and was not considered in our comparison. However, we did put the result of BNN (labeled as the ``Trained encoder+FC (BNN)" line) in Figure 5 and the accuracy drop between BNN to HDCs (BNN encoder+Majority Rule) is near-zero when the dimension is small.
>
>   2.1: It is true we can treat the rows in the Hadamard matrix which have strict orthogonality as hypervectors (with low dimension) to represent the value of each pixel. But if we use traditional HDC training methods, after a series of complex operations like binding, bundling, and majority rule, the final representation for each class that we used to do the classification will break the characteristics of the Hadamard matrix and orthogonality can no longer be guaranteed with low dimension. We performed the experiment suggested by the reviewer. On MNIST with $d=16$, both random sampling and the use of the Hadamard matrix failed to achieve good orthogonality in the final representation of each class. Using Equation 4, both had orthogonality values that were near 1 (which indicates almost no orthogonality) and the detection accuracy is a poor 10\%. However, when we use the BNN as an encoder as we had proposed, we found good orthogonality in the final representation, as shown in Figure 4. That is the reason for using the BNN as the encoder.
>
>   2.2: As shown in Figure 4, we evaluate the orthogonality of hypervectors across different thresholds. It can be seen with a proper threshold (between 500 to 5000), the orthogonality of hypervectors trained from BNN is good with a low dimensional ($d=16$).
>
>   2.3:  After carefully re-reading (Yu et al., 2022), we did not find any clear comparison of HDC performance with different orthogonalities. Orthogonality was only mentioned briefly in just four places in the paper.
>
>   2.4: The theoretical analysis in Section 3 illustrates the relationship between hypervectors dimensions and theoretical optimal accuracy. The unit separable ball assumption is a simplification for theoretical investigation purposes. Such kinds of assumptions are necessary to derive an explicit formula for worst-case accuracy. Relaxing the assumption to other distributions (box or polyhedron) might change the constant but the general trend should be the same. Also, this result is agnostic to the scale of the hypervectors. The general HDC may not satisfy this assumption but the proper encoder should still map the data into a bounded ball and each class should possess a certain pattern so that it can be classified by majority rule. Therefore the unit ball assumption is simple but still reasonable.
>
>   2.5: Yes, the high dimension still has certain benefits like robustness.
> We are trying to show that it is also possible to achieve good accuracy with a much lower hyper dimension, which significantly reduces the energy consumption during the inference stage.
>
> 3. The MNIST data to BNN is binary (either 0 or 1). For the input data larger than 0.2, we set it as 1. Otherwise, we set it as 0. Please see lines 87-97 and 341 in the codes "MNIST_sample.py".

---

> > ### Comment · Reviewer_jTeX · 2022-11-18
> > **Response**
> >
> > Thanks for the responses.
> >
> > 1. I agree the inference cost might be similar to standard HDC, however, the training cost is different as you'll train the binary neural network, currently there's no analysis of this training cost part in the paper, which make it an unfair comparison with some of existing baselines. Furthermore, also related to point 3, after checking through the code, your training of the encoder involves computations of floating numbers, the input is binarized, but the weights are not, such as the conv & batchnorm layer (you seem to quantize it after this floating computation), hence, your method will has similar training cost as a BNN. From figure 5, the result is clear that the accuracy performance is gained with training BNN but not from HDC, essentially you are just replacing the last FC layer of BNN with a HDC majority rule, and the training cost is nearly the same. This is not much to do with HDC or the low dimension advantage, but basically benefits from training a BNN.
> >
> > 2. I don't know where the claimed orthogonality is used? In standard HDC, we sampled uniformly from the high dimensional space to get item memory, which is used during encoding as $p_{f_i}$ in section 2. From my understanding, your claim is that this orthogonality can be found in low dimension with Hadamard matrix, then you can just go ahead with this method in section 2 as $p_{f_i}$ to encode (which I believe will perform badly). But now your claim seems to be the orthogonality of class representative vectors? Or even the encoded images? which is not claimed in standard HDC literature before to my knowledge. Can you explain where is the orthogonality you are claiming and where it's used? Also can you show this (near-)orthogonality in your BNN experiment to justify your claim? Otherwise still it looks to me the benefits come from training a BNN, but not from your (near-)orthogonality in low dimension.
> > -- 2.2 It's not clear how Figure 4 (Threshold study) shows the orthogonality, from my knowledge, the orthogonality is something the angle/inner product between pair of hypervectors? Figure 4 is the relation between threshold and some percentage (what is it, some accuracy?)
> >
> > 3. See above, the model weights for MNIST and CIFAR10 are floats, such as conv and Batchnorm layer.

---

> > > ### Author Response · Authors · 2022-11-19
> > > **Response**
> > >
> > >
> > > Thank you for your responses.
> > >
> > > 1. We agree that our training cost will be similar to the binary neural network and it is higher than traditional HDC with binding and bundling.
> > > 	1. However, training is a one-time affair while the models are used repeatedly in inference mode. Hence, as with most deep learning, training cost is not a major concern. The huge models used in natural language processing, for example, trains for extremely long time. In our case, the inference cost is in the same magnitude as standard HDC (matters of minutes), we feel that the comparison is fair.
> > > 	2. The BNN-type encoder is what we tried but it opens the door to other encoders which we suspect may do well too. We showed that good alternative encoders exist that can simplify and improve HDC while retaining its advantage - low cost in inference.
> > > 	3. The main point is that our encoder demonstrates the feasibility of representing the data in low dimensional ($d \approx 100$) space, which significantly improves HDC. Our Statements 3.1 and 3.2 also show the benefits of low-dimensional representation. The relationship between the low dimension and high accuracy is not a trivial conclusion that can be made from designing BNN variants. First, we prove the rationale for low-dimension cases using clustering methods like the majority rule. Next, we use dimension reduction and clustering methods to generate the representation of each digit. The similarity distance is used to perform the classification. It is not a simple, drop-in replacement of the FC layer with majority. The whole process is depicted in Figure 3.
> > >
> > > 2. First of all, the orthogonality refers to the orthogonality between class representative hypervectors, which is evaluated by Eq(3). It is the average of absolute cosine values for angles between the vector pairs. We have already included the explanation to the caption of Figure 4 to make it clearer.
> > > 	1. The orthogonality in Hadamard matrix is not directly used in our numerical experiments as we still require training to find a proper encoder. This is a result to aid the arguments of feasiblity of finding high orthogonality in the lower dimensional space. We are not claiming that using Hadamard matrix can give a low-dimensional class representative vector for HDC. Simply using vectors from Hadamard matrix instead of random sampling does not work.
> > > 	2. We evaluated the orthogonality (as defined in Eq(3)) for standard HDC and our model. The repeated numerical results show the standard HDC orthogonality is close to 1 (no orthogonality) when the dimension is low ($d=32, 64$).
> > > 	3. Our model is able to find low dimensional class representative vectors with a good orthogonality (see Figure 4). With a proper threshold, the orthogonality of representative vectors given by our model is about 0.3 for $d=32$. It can be seen that orthogonality and accuracy are correlated for different threshold values.
> > > 	4. (2.2) In Figure 4, the orthogonality is measured by the Eq (3): $\displaystyle \bar{O} = \frac{1}{K(K-1)} \sum_{c_1 \neq c_2} \frac{|\mathbf{R}^b_{c_1}*{\mathbf{R}^b_{c_2}}^T|}{d}$. This expression is the average of $\textrm{abs}(\cos(\textrm{angle}))$ of the class representative vector pairs.
> > >
> > > 3. We did a quick implementation of a variant for MNIST with integer weights. The results are in the newly added Appendix A.5. It can be seen that although integer weights is slightly worse than floating weights, it is still higher than other methods (https://anonymous.4open.science/r/supplementary-code-079B/d32_binary_weights.py, we kept batch normalization since compared with the millions of operations, the hundreds of operations in BN is insignificant.).

---

### Official Review · Reviewer_5WTt · 2022-10-26

**Confidence:** 2
**Correctness:** 3
**Technical Novelty And Significance:** 3
**Empirical Novelty And Significance:** 3
**Recommendation:** 6

**Clarity, Quality, Novelty And Reproducibility:**

This work is of decent quality, the idea and aspects look novel to the best of my knowledge. I have not run the code myself, but the code to reproduce is available in anonymous GitHub.

**Strength And Weaknesses:**

Strength:
* The paper is mostly well organized and is straightforward to follow.
* The study on dimension and accuracy looks very interesting. The result showing using much less dimensions looks very promising. I believe the technique proposed can be rather valuable especially for resource constrained hardware given the low dimension it can achieve.
Weaknesses:
* It would be helpful to provide more context on HDC basics, possibly some illustration figures to provide readers a more straight idea of how it works.
* Is it possible to have some more benchmarks on some other datasets other than MNIST? I understand that the authors have stated that the method needs some simple dataset, having some other datasets, maybe some simple classification datasets would be more convincing.
* Albeit having a better performance than existing methods, the performance on Cifar10 is not quite optimal from a application point of view. The author also states that HDC now mostly works on simple datasets.

**Summary Of The Paper:**

This paper studied the factors affecting accuracy in HDC and proposed a more efficient HDC method.

**Summary Of The Review:**

Overall, I think this is an important study to make HDC more applicable, despite having some limitations. The paper would benefit from more clarify on the background. More benchmarks, even on some other simple datasets, could make it more convincing.

---

> ### Author Response · Authors · 2022-11-10
> **Response to Reviewer 5WTt**
>
> Thank you for your suggestions, we have improved the writing for the background and added some illustration figures in the Appendix. We have also applied our methods to the Fashion-MNIST dataset and similar results were achieved (83.75\% and 85.25\% HDC accuracies with dimensions $d$ of 32 and 64 respectively).

---

### Official Review · Reviewer_781W · 2022-11-01

**Confidence:** 2
**Correctness:** 2
**Technical Novelty And Significance:** 3
**Empirical Novelty And Significance:** 2
**Recommendation:** 5

**Clarity, Quality, Novelty And Reproducibility:**

I list down the questions below:

1. What numerical simulation was performed in Figures 1 and 2 to get these two plots? Also, even though Statements 3.1 and 3.2 say the asymptotical degradation with respect to the hypervector dimension, how does the dimension close to 0 get the highest accuracy? Intuitively, some relatively large dimension is required to get the representational power of hypervectors (as the authors stated in $\epsilon$-quasiorthogonality).

2. What is the kernel-based binary encoder in Section 4? The current draft needs to explain in detail what this binary encoder is. For example, what is the weight sharing in Figure 3? What is the loss function to train the binary encoder?

3. In equation 2, the explanation on $w_{i, j}^l$ is not self-explanatory. The authors only say, "$i$,$j$ are the index of different neurons." I suspect $j$ is the index of neurons in $l-1$ layer, and $i$ is the neuron index at $l$th layer. Then, I am confused what $w_{i, j, x_{j}^{l-1}=1}^{l}$ means. It's never explained anywhere.

4. In equation 3, $G(x)$ is never defined anywhere, except mentioning that it's a gradient. I guess that the Jacobian is an identity matrix due to the straight-through estimator.

5. In Figure 5, the inference accuracy with the majority rule drops as the hypervector dimension goes up; however, the performance does not drop with the FC layer. Does this mean that performance degradation is not due to the dimensionality of the hypervector but due to the aggregation function, especially the majority rule?

6. The CIFAR-10 results, including the existing HDC methodology, seem to fail the learning procedure. CIFAR-10 can be achieved above 90% easily with ResNet architecture. From CIFAR-10 results, I am skeptical of MNIST results because the effective dimensionality of MNIST is less than 768 [1]; therefore, low-dimension hypervectors work well. Furthermore, the bad performances of overall HDC methods in CIFAR-10 worry me about whether the current HDC framework is a promising direction for the ML community.

[1] Bubeck, Sébastien, and Mark Sellke. "A universal law of robustness via isoperimetry." Advances in Neural Information Processing Systems 34 (2021): 28811-28822.

**Strength And Weaknesses:**

Strength

1. A theoretical framework to explain the relationship between the hypervector dimensions and model performance.
2. The author's empirical results are better than the existing methodology.

Weakness
1. Hard to read.
2. Notations are not well defined. Mainly, I could not follow the idea in the Methods section.
3. Figures are not self-explanatory. Hard to understand what idea is conveying.
4. HDC method seems to only work on MNIST dataset. CIFAR-10 results are overall very poor.

**Summary Of The Paper:**

This paper proposes a new hyperdimensional computing (HDC) methodology with low-dimensional hypervectors. The authors provide the theoretical results (for a binary classifier case) that the relatively lower dimensions can yield better performance. Furthermore, their theories suggest that the model performance asymptotically drops with respect to the hypervector dimension. Finally, the author empirically shows that their HDC method performs better than the existing methodology.

**Summary Of The Review:**

The authors provide theoretical and empirical results regarding the new hyperdimensional computing framework to accelerate the inference speed. However, overall the paper could be clearer to read and more self-explanatory. Mainly, I could not understand what methodology they used as their algorithm. The one big concern is that this HDC method works well on MNIST but not other datasets (CIFAR-10), which alerts my internal false alarm. Overall, I do not think the draft does not meet the ICLR bars.

---

> ### Author Response · Authors · 2022-11-10
> **Response to Reviewer 781W**
>
> 1.  Figure 1 was drawn directly using the exact mathematical expression of $Acc^w_{2, d}$. For Figure 2, we were unable to find the exact expression for $\bar{Acc}\_{K, d}$. Hence, we used Monte Carlo simulation sampling $\theta_1$ and $\theta_2$ 1000 times to evaluate the expectation form $E_{\theta_1, \theta_2 \sim U[0, 1]^d} \bigg [ 1 - \frac{\arccos \big( \sup_{j=1}^d \frac{\sum_{i=1}^j |\theta_1 - \theta_2|_{(i)}}{\sqrt{j} \|\theta_1 - \theta_2\|} \big )}{\pi} \bigg ]$. We will add the details in the new version. Thank you for pointing it out. In Statements 3.1 and 3.2, the theoretical results we derived are based on the assumption that we have the ''optimal'' majority rule, i.e., the optimal $\hat{\theta}_1, \dots, \hat{\theta}_K$.
> However, in practice, it was observed that under certain circumstances lower dimensions did not give better performance because we are unable to use the optimal majority rule.
> For general $K$-classes classification, it is also true we need a relatively large dimension $d$, i.e. $d > log_2 K$.
> We mentioned it in the very beginning of the paper:  ''In particular, if the hypervector dimension $d$ is sufficient to represent a vector with $K$ classes ($d > \log_2 K$) then, the lower the dimension, the higher the theoretical optimal accuracy." Also, it is true that intuitively we may need larger dimensions to improve the representational power of the hypervectors. This is why HDC works use hypervectors of thousands of dimensions to represent the data. We had mentioned it in the introduction: ``It is intuitively true that high dimensions will lead to high orthogonality and representational power, contrary to popular belief, we found that as the dimension of the hypervectors $d$ increases, the upper bound for inference accuracy actually decreases (Statement 3.1 and Statement 3.2)." The main contribution of this paper is in finding the limit of the dimensions and in doing so exposing the factors influencing that number.
>
> 2. For weight sharing, we have explained in the workflow described at the beginning of section 4: ``we first combine a kernel-based binary encoder with a fully-connected layer and train the whole network. Next, the fully-connected layer is replaced with a majority rule." `Weight sharing' means that we use the same weights before and after we replaced the FC layer with the majority rule. The loss function used for training is cross-entropy loss. We added a detailed explanation of this and what loss function we used (the exact expression) in the new version. Thank you for pointing it out. The description of the structure and workflow of our kernel-based binary encoder in Section 4.1 is also improved.
>
> 3. The meaning of $w_{i,j}^l$ is given in Section 4.1: ''where $x^l_i$ indicates the output of layer $l$ at neuron $i$, $ w^l_{i,j}$ and $ b^l_i$ indicates the weight and bias at layer $l$ ($i,j$ are the index of different neurons)." But we will add a more detailed explanation as you recommended ($j$ is the index of neurons in $l-1$ layer, and $i$ is the neuron index at $l$ th layer.). For the meaning of $w_{i,j,x_{l-1}=1}$, we explained it right above the equations: ``We only need to sum up the weights whose corresponding inputs are 1. Weights whose corresponding inputs are 0 can be ignored." The meaning of $x_{l-1}$ and $w_{i,j}$ is also explained. However, we will also add a more detailed explanation: $w_{i,j,x_{l-1}=1}$ indicates the weight whose corresponding inputs $x_{l-1}$ are 1.
>
> 4.  $G(x)$ is the gradient for backpropagation. Because the whole function is discontinuous and not differentiable at the turning points, we use the method of the straight-through estimator to simulate the gradient. The gradient is set as 1 to make the whole network trainable. Thank you for pointing it out. We will add the definition to the manuscript.
>
> 5. Yes, all our statements, assumptions, and proof are in the HDC domains where aggregation functions such as the majority rule are used for classification. The fully connected layer is merely an intermediary to obtain a good encoder. The former is not directly used in the HDC model.
>
> 6. As with all tools, HDC do have limitations. In particular, it cannot achieve very high accuracy on some complex dataset. We discuss such limitations in Section 6. However, HDC has its advantages compared with other machine learning algorithms.
>  Low energy consumption (low latency) and high parallelism make HDCs well-suited for resource-constrained use-cases. We discussed several application scenarios of HDCs in the introduction section. Just as supercomputers do not negate the role of the humble Raspberry Pi used in embedded devices, we believe there is a role for tools of different capabilities.

---

> > ### Comment · Reviewer_781W · 2022-11-17
> > **Response**
> >
> > 1. Thanks for the explanation; however, I am still not convinced about the results of Figures 1 and 2. The authors mention in the rebuttal that "Also, it is true that intuitively we may need larger dimensions to improve the representational power of the hypervectors," while statements 3.1 and 3.2 tell us the upper bound of inference accuracy decreases in asymptotical perspective. Then, Figures 1 and 2 should look like bell-shaped functions, according to my understanding.
> >
> > 2. Thanks for pointing out the weight-sharing definition. In this case, weight-sharing is not the proper terminology; weight-sharing is a popular term in neural architecture search [1, 2] or convolutional kernels in CNN. I suggest changing to transferring the weights instead.
> >
> > 3. I don't see the necessity of replacing the multiplications with the additions. In BNN literature, matrix multiplications are replaced with XNOR and BITCOUNT operations, which is very efficient. Then, the sign function makes the multiplication output -1 or 1.
> >
> > 4. N/A
> >
> > 5. Thank you for the explanations.
> >
> > 6. I can't entirely agree with the authors. Binary neural networks [3] can achieve better results than the HDC frameworks achieving higher CIFAR-10 test accuracy (>80%). Although the authors achieved a better result than the existing HDC method, I am still skeptical about the methodology since it only works on the MNIST dataset.
> >
> > My stance is the same as before. Therefore, I will keep my score.
> >
> > [1] Xie, Lingxi, et al. "Weight-sharing neural architecture search: A battle to shrink the optimization gap." ACM Computing Surveys (CSUR) 54.9 (2021): 1-37.
> > [2] Li, Liam, and Ameet Talwalkar. "Random search and reproducibility for neural architecture search." Uncertainty in artificial intelligence. PMLR, 2020.
> > [3] Courbariaux, Matthieu, et al. "Binarized neural networks: Training deep neural networks with weights and activations constrained to+ 1 or-1." arXiv preprint arXiv:1602.02830 (2016).

---

> > > ### Author Response · Authors · 2022-11-18
> > > **Response**
> > >
> > >
> > > Thank you for your response.
> > >
> > >  1. The theoretical results we derived are based on the optimal assumption. We show the whole process of proof in the Appendix (A.2 LEMMA AND PROOF)
> > >  2. We will change it to “transferring the weights”. Thank you for pointing this out.
> > >  3. As far as we have checked in the GitHub sources of BNNs, matrix multiplication is required at least in the first layer to binarize the input.[1] Also, batch normalization is always used which may take additional multiplications.
> > > 4. N/A
> > > 5. N/A
> > >  6. The subject of the paper is the limit of HDC – not the merits of BNN or any other model. As stated, we believe that every tool has its use-scenario [2,3,4] and we are not alone – there is sufficient literature on HDC to show that there is significant interest on HDC as a model. The goal of our paper is to aid the HDC community in improving HDC."
> > >
> > > [1] Hubara, et al. "Binarized neural networks." Advances in neural information processing systems, 29, 2016
> > >
> > > [2]Manuel Schmuck, Luca Benini, and Abbas Rahimi. Hardware optimizations of dense binary hyperdimensional computing: Rematerialization of hypervectors, binarized bundling, and combi- national associative memory. ACM Journal on Emerging Technologies in Computing Systems (JETC), 15(4):1–25, 2019.
> > >
> > > [3]Peer Neubert, SteKanervafan Schubert, and Peter Protzel. An introduction to hyperdimensional computing for robotics. KI-Ku ̈nstliche Intelligenz, 33(4):319–330, 2019.
> > >
> > > [4]Abbas Rahimi, Pentti Kanerva, and Jan M Rabaey. A robust and energy-efficient classifier using brain-inspired hyperdimensional computing. In Proceedings of the 2016 international symposium on low power electronics and design, pp. 64–69, 2016.

---

> > > > ### Comment · Reviewer_781W · 2022-11-18
> > > > **Response**
> > > >
> > > > Thank you for your response.
> > > >
> > > > 1. According to the assumption, the authors assume that hypervectors are uniformly distributed in the $d$-dimensional l2-ball. The problem I see here is that data complexity ($d$) is correlated to the dimensionality of $\theta$. The proper setup should be a given the hypervectors from fixed $m$-dimensional l2-ball, then study the worst-case or average accuracy of $d$-dimensional support hypervector $\theta$, where the data should be linearly projected to $d$ dimensional vectors. The point is that the complexity of the hypervector changes as the dimensionality of $\theta$ changes as well. I believe this is the main reason for getting Figures 1 and 2 plots.
> > > > 2. N/A
> > > > 3. The point is that I do not see Eq 2 as one of the contributions from the author. XNOR, BITCOUNT, and SIGN activation functions can efficiently compute binary matrix multiplications.
> > > > 4. N/A
> > > > 5. N/A
> > > > 6. The fact that the methodology only works on MNIST is a false alarm to me, but I understood the author's point. Thank you.

---

> > > > > ### Author Response · Authors · 2022-11-18
> > > > > **Response**
> > > > >
> > > > > Thank you for your response.
> > > > >
> > > > > 1. In terms of setup comparison, our model is assuming that the encoder can represent the raw data into a linearly-separable unit ball. This is a reasonable and feasible assumption for the data dimension $m$ sufficiently large. As we mentioned in the paper, we should have constraints such as $m \geq \log_2 K$.
> > > > > 	1. The setup you proposed is from another perspective, which assumes the underlying dimension for data is exactly $m$.
> > > > > 	2. First, considering the region $d < m$, the result for $d$-dimensional majority rule is equivalent to randomly setting $m-d$ coordinates of $\hat{\theta}$ to 0 (if the projection is done coordinate-wise) and optimizing over the rest $d$ coordinates. The low-dimensional case will suffer from both misclassification and misrepresentation. The misclassification error refers to the error in our model, while the misrepresentation error refers to the error induced by the discrepancy between the projected distribution and $d$-dimensional uniform unit ball distribution. The misclassification error, according to our model, should be small but the misrepresentation error can be much larger as $d \to 1$. This trend is compatible with the left part of the accuracy curve in our numerical experiments. Then the accuracy will increase over $d < m$ region.  Different linear projection schemes should give similar results.
> > > > > 	3. The worst-case $K$-classes prediction accuracy over $m$-dimensional data projected to $d$-dimensional subspace is $Acc^w\_{K, m, d} := \inf\_{\theta\_1, \theta\_2, \dots, \theta\_K \in [0, 1]^m} \sup\_{co\_1, \dots, co\_d} \sup\_{\hat{\theta}\_1, \hat{\theta}\_2, \dots, \hat{\theta}\_K \in \Theta\_{co\_1, \dots, co\_d}} \mathbb{E}\_x \bigg [ \sum\_{i=1}^K \prod\_{j \neq i} \mathbf{1}\_{\\{\theta\_i \cdot x > \theta\_j \cdot x\\}} \mathbf{1}\_{\\{\hat{\theta}\_i \cdot x > \hat{\theta}\_j \cdot x\\}} \bigg ] \leq Acc^w\_{K, m, d+1}  \leq Acc^w\_{K, m}$, where $\Theta\_{co\_1, \dots, co\_d} = \\{ \theta | \theta\_{i} \in \\{0, 1\\}, i \in \\{co\_1, \dots, co\_d\\}; \theta\_{i} = 0, i \not \in \\{co\_1, \dots, co\_d\\} \\}$. It can be seen that in this setup the accuracy is increasing over the region $d<m$.
> > > > > 	4. Moreover, if we consider the region for $d > m$, the accuracy in this setup will be constant. These two regions (both $d<m$ and $d>m$) imply the same idea as our statements: with hyperdimension $d$ sufficiently large, there is no benefit to using an exceedingly larger hyperdimension.
> > > > > 	5. Therefore, this new setup does not contradict our result as we do observe the low accuracy for dimension $d \leq 64$ in numerical experiments (Figure 5 and Figure 6). We thank the reviewer as this is a useful setup to aid the explanation for the left component of the accuracy curve. We have incorporated this useful insight into the paper (See Appendix A.4).
> > > > > 2. N/A
> > > > > 3. Equation 2 describes what we did for binarization. We have not explored other ways of doing it (using XOR etc) but it is orthogonal - a more efficient way to compute this binarization will benefit the overall model. We are not claiming Eq 2 as our contribution, this treatment is a common practice in the HDC domain.
> > > > > 4. N/A
> > > > > 5. N/A
> > > > > 6. N/A.

---

> > > > > > ### Comment · Reviewer_781W · 2022-11-18
> > > > > > **Response**
> > > > > >
> > > > > > Thank you for the response. I now see the point of claims from the authors. I suggest authors make each figure caption with detailed descriptions that the figure is self-explanatory.
> > > > > >
> > > > > > I raise my score to 6; however, my skepticism remains the same that the methodology only works on MNIST. Thanks to the authors for the discussions that gave me a better understanding of this paper.

---

> > > > > > > ### Comment · Reviewer_781W · 2022-12-04
> > > > > > > **Revising the score**
> > > > > > >
> > > > > > > After reconsidering the current draft and other reviewers' responses, I revise my score to 5. The concerns about the gap between theory and experiments still remain:
> > > > > > > 1. BNN counteracts the computational benefit of hypervectors, and Binary Neural Network does not guarantee quasi-orthogonality
> > > > > > > 2. Experimental results only work on MNIST-like datasets.

---

> > > > > > > > ### Author Response · Authors · 2022-12-05
> > > > > > > > **Response**
> > > > > > > >
> > > > > > > > Thank you for your comments.  In the paper, we showed theoretically in statements 3.1, 3.2, and 3.3 that the use of lower dimensions can also achieve equal or even higher detection accuracy compared with other state-of-the-art HDC models. We then verified them in our experiments, achieving state-of-the-art HDC accuracies with a dimension of 100 or less.
> > > > > > > >
> > > > > > > > For the first point, using BNN as an encoder, our proposal uses even fewer encoder additions and Boolean operations than traditional HDC methods. The compute operations during inference are now a tenth that of traditional HDCs.  For the similarity checking, since the number of operations in computing the Hamming distance is linearly proportional to the dimension of the hypervectors. Hence, our method only needs 0.32% of the operations required by HDC models using a dimension of 10,000.
> > > > > > > > Also, as we showed in Figure 4, the hypervectors after the BNN encoder stage have good orthogonality. For the same dimension, the orthogonality measure of traditional HDCs is near 1 going by Equation 4, which translates to almost no orthogonality, resulting in a detection accuracy of around 10%. As you suggest, we modified the titles of our figures with detailed descriptions in the latest version. Thank you for your suggestion.
> > > > > > > >
> > > > > > > > For the second point, it is true that most HDC methods cannot achieve good accuracy on complex datasets like CIFAR-10 – for the moment. However, using our methods, we achieve state-of-art HDC results on CIFAR-10. With a dimension of only 128, we achieved an HDC accuracy of 46.18%, which is higher than the state-of-the-art HDC models. As we have stated in our other comments, we believe that while HDC may not be able to handle complex datasets as with other cloud-based models, it is a promising technology for applications with properly scoped use cases in edge AI due to its energy efficiency. We believe that the insights of our paper pave the way for significantly improved HDC models.

---

### Author Response · Authors · 2022-11-10
**Summary of Responses**

This paper aims to investigate the limits of hyperdimensional computing (HDC). We are not trying to belittle it because we believe that different tools are needed for different usage scenarios and HDC has its role in edge and resource-constrained deployment scenarios, even if it is unable to handle very large tasks. We believe that the insights of this paper will help improve HDC in the future. With that in mind, we would like to respond to the reviewers' comments.

---

### Public Comment · ~Shijin_Duan1 · 2023-02-09
**A Comment On HDC Dimension Reduction Topic**

Glad to see a work improving HDC performance! We are also researching similar topics, and recently published our findings in tinyML'22 [1], in which we successfully reduced the dimensions of HDC by orders of magnitude (e.g., 4/64 for MNIST dataset), yet why we name it as Low Dimensional Computing or LDC. Specially, our LDC projects the classic HDC process to BNN structure, and it can retain all efficiency that HDC has during inference, as well as the accuracy. You are welcome to discuss any considerations with us.

[1] Shijin Duan, Xiaolin Xu, and Shaolei Ren. A brain-inspired low-dimensional computing classifier for inference on tiny devices. In tinyML Research Symposium 2022, 2022.

---

### Decision · Program_Chairs · 2023-01-20

**Decision:**

Reject

**Justification For Why Not Higher Score:**

As described in the summary and the notes from the PC meeting, far too many concerns still remain to merit acceptance at this stage.

**Justification For Why Not Lower Score:**

N/A

**Metareview: Summary, Strengths And Weaknesses:**

The paper proposes a new approach in hyperdimensional computing (HDC), which is an emerging model of efficient computation involving solely binary representations (called hypervectors) for data. Typically, these hypervectors are chosen to be _random_ and _very high-dimensional_; this paper theoretically argues that lower dimensional representations suffice, provided that the hypervectors are quasi-orthogonal. Additionally, the paper proposes using a binary neural network (BNN) to derive hypervectors of very low dimension. Results on (Fashion-)MNIST shows the benefits of the proposed approach.

Original reviews for this paper were borderline. While the reviewers acknowledged the importance of the problem and the originality of the proposed ideas, several other concerns were raised. The use of BNNs as feature encoders seemed contrary to the spirit of HDC. The primary advantage for using HDC is a dramatic reduction in computation cost compared to standard floating point arithmetic, but training BNNs incurs additional costs and negates any such advantages; in any case, an apples-to-apples comparison here would have to take into account BNN training costs, which the authors did not provide.

The theoretical results on quasi-orthogonality assumed that hypervectors were randomly sampled, but the experimental results discarded this approach and used a BNN encoder (trained offline). Finally, the fact that the only results were on (Fashion-)MNIST raised questions about broader impact. Responses by the authors did not significantly affect the reviewers' opinions.

Overall, on balance the paper falls short of the bar for acceptance. The authors might consider (for a future revision) better justifying the use of BNNs as a sound strategy, tightening the connection between theory and practice, as well as clearing any doubts about future impact by performing more challenging experiments.



**Summary Of Ac-Reviewer Meeting:**

The virtual PC meeting was held on Dec 2 over Zoom to discuss this paper. Two reviewers and the AC attended this meeting; the third reviewer corresponded with the AC later via email.

Broadly speaking, concerns about this paper remained. There seems to be a fairly wide disconnect between theory and experiments: the theory talked about quasi-orthogonality, while the experiments used a BNN encoder to obtain hypervectors. The latter was a particularly major concern, since the use of BNNs negates much of the computational benefits of using hypervectors (previous approaches simply used random features). There were also concerns about broad applicability and impact of the proposed method, since the only empirical validation was on (Fashion-)MNIST. The reviewers were requested to consider updating their scores keeping in mind this discussion, each others' reviews, and the authors' responses.

The point about using BNNs, as well as the wide disconnect between the theory and experiments in the paper, were the deciding factors in my final decision.